



# Beaching patterns of plastic debris along the Indian Ocean rim

Mirjam van der Mheen[1], Erik van Sebille[2], and Charitha Pattiaratchi[1]

[1]Oceans Graduate School and the UWA Oceans Institute, the University of Western Australia, Perth, Australia
[2]Institute for Marine and Atmospheric Research Utrecht, Utrecht University, Utrecht, the Netherlands

**Correspondence:** Mirjam van der Mheen (mirjam.vandermheen@research.uwa.edu.au)

**Abstract.** A large percentage of global ocean plastic waste enters the northern hemisphere Indian Ocean (NIO). Despite this, it is unclear what happens to buoyant plastics in the NIO. Because the subtropics in the NIO is blocked by landmass, there is no subtropical gyre and no associated subtropical garbage patch in this region. We therefore hypothesise that plastics "beach" and end up on coastlines along the Indian Ocean rim. In this paper, we determine the influence of beaching plastics by applying

different beaching conditions to Lagrangian particle tracking simulation results. Our results show that a large amount of plastic likely ends up on coastlines in the NIO, while some crosses the equator into the southern hemisphere Indian Ocean (SIO). In the NIO, the transport of plastics is dominated by seasonally reversing monsoonal currents, which transport plastics back and forth between the Arabian Sea and the Bay of Bengal. All buoyant plastic material in this region beaches within a few years in our simulations. Countries bordering the Bay of Bengal are particularly heavily affected by plastics beaching on coastlines. This

is a result of both the large sources of plastic waste in the region, as well as ocean dynamics which concentrate plastics in the Bay of Bengal. During the intermonsoon period following the southwest monsoon season (September, October, November), plastics can cross the equator on the eastern side of the NIO basin into the SIO. Plastics that escape from the NIO into the SIO beach on eastern African coastlines and islands in the SIO or enter the subtropical SIO garbage patch.

## 1 Introduction

Large amounts of plastic waste enter the ocean every year (Jambeck et al., 2015; Lebreton et al., 2017; Schmidt et al., 2017), potentially harming marine species and ecosystems (Law, 2017). A large percentage of global plastic waste is estimated to enter the Indian Ocean. Despite this, buoyant marine plastic debris ("plastics") is relatively under-sampled and under-studied in the Indian Ocean (van Sebille et al., 2015). The Indian Ocean atmospheric and oceanic dynamics are unique (Schott et al., 2009), so the dynamics of plastics in the Indian Ocean differ from those in the other oceans (van der Mheen et al., 2019).

In the Pacific and Atlantic oceans, plastics accumulate in so-called "garbage patches" in the subtropical ocean gyres (e.g. Moore et al., 2001; Maximenko et al., 2012; van Sebille et al., 2012; Lebreton et al., 2012; Eriksen et al., 2013; van Sebille et al., 2015). Plastics also accumulate in a subtropical garbage patch in the southern hemisphere Indian Ocean, but it is much more dispersive and sensitive to different transport mechanisms (currents, wind, waves) than the garbage patches in the other oceans (van der Mheen et al., 2019). In contrast, the subtropical northern hemisphere Indian Ocean is blocked by landmass, so

there is no subtropical gyre and associated garbage patch. In addition, it is unclear if plastics entering the northern hemisphere



Indian Ocean cross the equator into the subtropical garbage patch in the southern hemisphere, as we explain further in the following paragraphs.

Strong currents are known to act as transport barriers for buoyant objects (McAdam and van Sebille, 2018). For example, most fluid parcels in the Gulf Stream flow downstream; cross-stream transport only occurs at depth (Bower, 1991). As a result,

there is almost no surface transport between the subtropics and the subpolar region in the North Atlantic Ocean: in 30 years only one ocean surface drifter crossed this boundary (Brambilla and Talley, 2006). In the equatorial region, the easterly trade winds drive strong equatorial currents and counter-currents (Dijkstra, 2008). As a result, ocean surface drifters do not tend to cross the equator and ultimately return to their original hemisphere (Maximenko et al., 2012). It has therefore been suggested that plastics do not generally cross the equator but remain in the hemisphere where they entered the ocean (Lebreton et al.,

35 2012).

However, in contrast to the other oceans, the easterly trade winds in the northern hemisphere Indian Ocean are not steady. Instead, they generally only have an easterly component during December, January, and February and have a westerly component during the remainder of the year (Schott et al., 2009). As a result, the North Equatorial Current and the South Equatorial Counter Current in the Indian Ocean are not steady either. In addition, although the surface connectivity is split into two

hemispheres in both the Pacific and Atlantic oceans, the surface of the Indian Ocean appears connected between hemispheres (Froyland et al., 2014). Because of this, it is unclear if plastics tend to remain in their original hemisphere in the Indian Ocean. The question is therefore what happens to plastics entering the northern hemisphere Indian Ocean (NIO).

Measurements of open ocean plastic concentrations in the Indian Ocean are scarce (Figure 1; van Sebille et al., 2015) and insufficient to determine the fate of plastics entering the NIO. However, numerical modelling studies show a garbage patch

forming in the Bay of Bengal (Lebreton et al., 2012; van der Mheen et al., 2019). Sampling studies confirm that there are high concentrations of plastics in the Bay of Bengal (Ryan, 2013), but it is not clear whether this is a result of plastics accumulating here or due to large nearby sources.

Another hypothesis is that plastics end up on coastlines in the NIO. Multiple studies sampled plastics on beaches in the Indian Ocean (Figure 1; Ryan, 1987; Slip and Burton, 1991; Madzena and Lasiak, 1997; Uneputty and Evans, 1997; Barnes,

2004; Jayasiri et al., 2013; Duhec et al., 2015; Nel and Froneman, 2015; Bouwman et al., 2016; Kumar et al., 2016; Imhof et al., 2017; Lavers et al., 2019), but because they used very different sampling methods on different timescales (Table A1), their results are difficult to compare. However, they do provide qualitative evidence that plastic is found on coastlines throughout the Indian Ocean, both on populated beaches close to plastic sources (Uneputty and Evans, 1997; Jayasiri et al., 2013; Kumar et al., 2016) as well as on remote, uninhabited coastlines and islands (Ryan, 1987; Slip and Burton, 1991; Madzena and Lasiak,

1997; Barnes, 2004; Duhec et al., 2015; Nel and Froneman, 2015; Bouwman et al., 2016; Imhof et al., 2017; Lavers et al., 2019). Which coastlines are most heavily affected by stranding plastics depends both on the location of plastic sources and the ocean dynamics in the region.

In the NIO, both the atmospheric and oceanic dynamics are dominated by the monsoon system, which is driven by differences in air temperature above the Asian continent and above the NIO (Schott et al., 2009). During the southwest monsoon season

(boreal summer: June, July, August) the air above the Asian continent is warmer than above the ocean, leading to predominantly





south-westerly winds. In contrast, during the northeast monsoon season (boreal winter: December, January, February) the air above the ocean is warmer than above the Asian continent, resulting in predominantly north-easterly winds. These monsoonal winds result in strong seasonal variations in ocean surface currents in the NIO and Indian Ocean equatorial region.

During the northeast monsoon season the Northeast Monsoon Current (NMC) flows from the Bay of Bengal westwards past Sri Lanka and into the Arabian Sea (Figure 1a; Schott et al., 2009; de Vos et al., 2014). The North Equatorial Current (NEC) also flows westwards during this season, feeding into the south-westward flowing Somali Current (SC), which in turn feeds into the eastwards South Equatorial Counter Current (SECC). The South Java Current (SJC) flows south-eastwards along Sumatra and Java, but is relatively weak during the northeast monsoon season (Sprintall et al., 2010).

During the southwest monsoon season the NMC dissolves and instead the Southwest Monsoon Current (SMC) flows from the Arabian Sea eastwards past Sri Lanka and into the Bay of Bengal (Figure 1b; Schott et al., 2009; de Vos et al., 2014). There is no NEC during this season, and as a result the SC reverses direction as it is supplied by the westward flowing South Equatorial Current (SEC) and the East African Coastal Current (EACC). The SJC continues to flow south-eastwards along Sumatra, but flows north-westwards along Java (Sprintall et al., 2010) as it is supplied by the strengthening Indonesian Throughflow (ITF) during the southwest monsoon season (Sprintall et al., 2009). At the convergence of the two opposing flows, a current flows south-westwards and feeds into the SEC.

During the intermonsoon periods strong eastward flowing surface Wrytki jets develop along the equator (Wyrtki, 1973), which are unique to the Indian Ocean. The Wrytki jets are strongest during the intermonsoon period following the southwest monsoon season (Qui and Yu, 2009). They strengthen the SJC, which flows south-eastwards during the intermonsoon periods.

The aim of this paper is to determine how these seasonally reversing ocean surface currents transport plastics that enter the NIO. Specifically, we focus on which coastlines are most heavily affected by stranding plastics. For convenience, we refer to plastics stranding on coastlines as "beaching" or "beached plastics", where beaching can occur on any type of coastline, not just beaches. In addition to surface currents, wind and waves have a significant impact on the dynamics of buoyant objects in the southern hemisphere Indian Ocean (SIO; van der Mheen et al., 2019). However, we only consider the influence of surface currents on the transport of plastics in this study; including dynamics due to wind and waves are beyond the scope of this paper. We discuss the reasons behind this as well as the possible implications in more detail in section 4.

Our results show that plastics in the NIO move back and forth between the Bay of Bengal and the Arabian Sea, following monsoonal winds and currents. Plastics beach on coastlines throughout the NIO. Countries bordering the Bay of Bengal are most heavily and consistently affected. We also show that plastics from the NIO can cross the equator into the SIO. In our simulations, this mainly occurs during the intermonsoon period following the southwest monsoon season (September, October, November), and we suggest a mechanism for the "escape" of plastics from the NIO into the SIO. Plastics that cross into the SIO beach along the entire eastern African coastline as well as on remote islands.

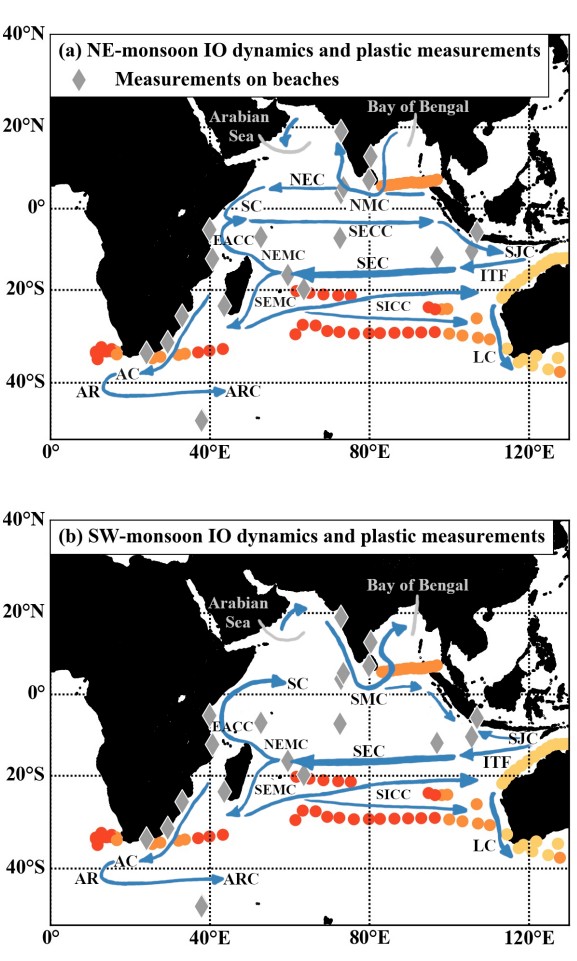

**Figure 1.** Overview of standardised open ocean plastic measurements in the Indian Ocean (filled circles); approximate locations of sampling studies of plastics on beaches (grey diamonds); and schematic dominant ocean surface currents (blue arrows) during the (a) northeast monsoon season; and (b) southwest monsoon season. Open ocean sampling studies were performed by Morris (1980); Reisser et al. (2013); Eriksen et al. (2014); Cózar et al. (2014) and standardised by van Sebille et al. (2015). Sampling studies of plastics on beaches were performed by Ryan (1987); Slip and Burton (1991); Madzena and Lasiak (1997); Uneputty and Evans (1997); Barnes (2004); Jayasiri et al. (2013); Duhec et al. (2015); Nel and Froneman (2015); Bouwman et al. (2016); Kumar et al. (2016); Imhof et al. (2017); Lavers et al. (2019). Schematic ocean surface currents are based on Schott et al. (2009). The following currents are shown and labeled with their abbreviations: Northeast Monsoon Current (NMC) and Southwest Monsoon Current (SMC); North Equatorial Current (NEC); Somali Current (SC); South Equatorial Counter Current (SECC); South Java Current (SJC); East African Coastal Current (EACC); Indonesian Throughflow (ITF); Northeast Madagascar Current (NEMC); Southeast Madagascar Current (SEMC); Agulhas Current (AC); Agulhas Retroflection (AR); Agulhas Return Current (ARC); South Indian Counter Current (SICC); Leeuwin Current (LC).





## 2 Methodology

### 2.1 Plastic sources

Global plastic waste inputs from coastlines were estimated by Jambeck et al. (2015), and inputs from rivers were estimated by

both Lebreton et al. (2017) and Schmidt et al. (2017). The estimate by Jambeck et al. (2015) is based on a fixed percentage of

mismanaged plastic waste per country entering the ocean. In addition to mismanaged plastic waste, Lebreton et al. (2017) and

Schmidt et al. (2017) included the influence of river catchment geography and river discharge to estimate how much plastic

waste enters the ocean. They also calibrated their estimates based on available measurements of plastic concentrations in rivers

around the globe. The total amount of plastic waste entering the ocean from rivers each year estimated by Lebreton et al. (2017)

and Schmidt et al. (2017) agree relatively well with each other. In contrast, the estimate by Jambeck et al. (2015) is an order

of magnitude larger. In this paper, we use plastic waste input from rivers estimated by Lebreton et al. (2017) as plastic source

locations in our simulations (section 2.2). These inputs are based on measurements of floating plastics in rivers with size ranges

between 0.3 mm and 0.5 m, and are the more conservative option compared to those of Jambeck et al. (2015).

The largest plastic source locations in the NIO are located around the Bay of Bengal and on the eastern side of the Arabian

Sea (Figure 2a). Lebreton et al. (2017) derived monthly plastic waste inputs, which mainly vary depending on river discharge.

The wet season with the largest discharges is in boreal summer in the NIO, and plastic waste input in the region peaks in

August (Figure 2b).

### 2.2 Particle tracking simulations

We use OceanParcels-v2 (Lange and van Sebille, 2017; Delandmeter and van Sebille, 2019) to run Lagrangian particle tracking

simulations of plastics released in the NIO, forced by ocean surface currents from HYCOM+NCODA Global 1/12° Reanalysis

data ("HYCOM"; Cummings, 2005; Cummings and Smedstad, 2013). Ocean surface currents from HYCOM are available at

3 hourly temporal resolution and 1/12° horizontal resolution from 01-01-1995 to 31-12-2015. We use a timestep of $dt = 1$

hour in the particle tracking simulations and use 5 day outputs of particle locations for analysis. We include Brownian particle

diffusion with a constant horizontal diffusion coefficient of $K_h = 10.0 \ \mathrm{m^2 s^{-1}}$. We determined the value of $K_h$ following the

definition of Peliz et al. (2007): $K_h = \epsilon^{1/3} dx^{4/3}$, where $\epsilon = 10^{-9} \ \mathrm{m^2 s^{-3}}$ is the turbulent dissipation rate, and $dx = \mathcal{O}(10) \ \mathrm{km}$

is the size of a grid cell in HYCOM.

We limit the domain of our particle tracking simulations between 0° E to 130° E, and 50° S to 40° N. Particles are removed

from the simulation after passing through these boundaries. We choose this relatively large domain because we are interested

in the amount of particles that cross from the NIO into the SIO, as well as any particles that escape from the SIO into the other

ocean basins. The simulation domain extends relatively far east and south to include the Agulhas retroflection (e.g. Gordon,

2003), so that any particles caught in the retroflection can "escape" from the SIO into the South Atlantic Ocean, but also

potentially move back into the SIO with the Agulhas return current. The definitions of the NIO and SIO that we use are shown

in Figure A1.



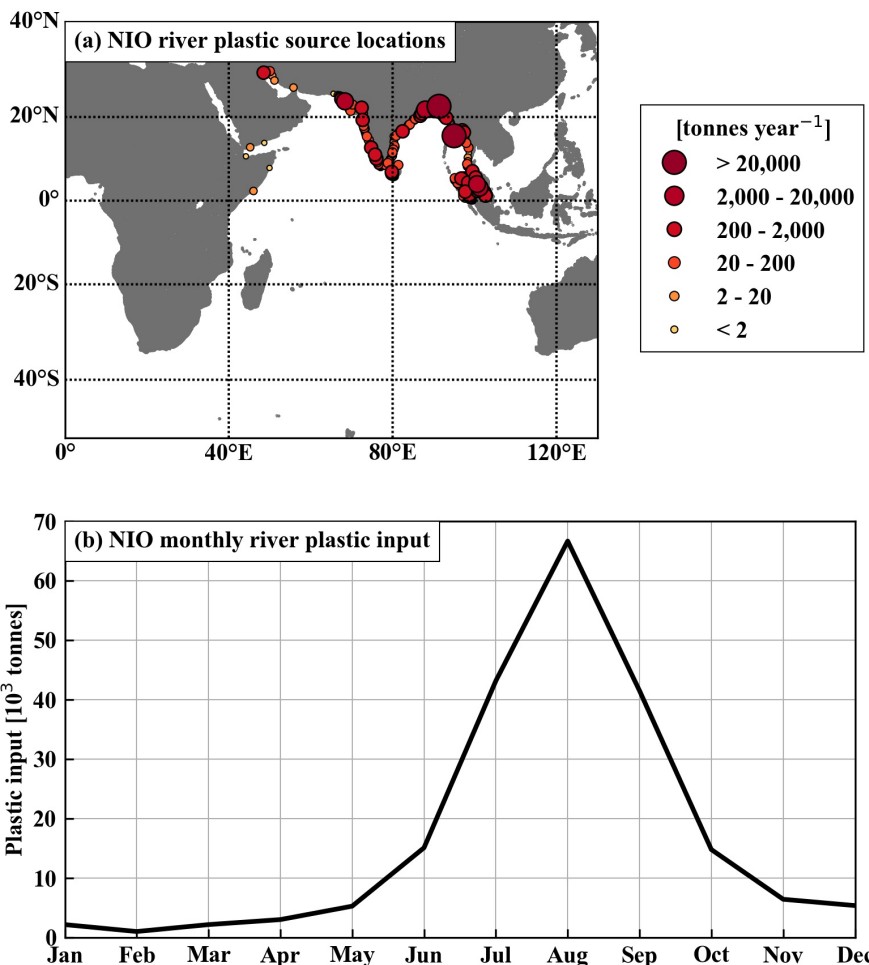

**Figure 2.** (a) Locations of plastic waste input from rivers in the northern hemisphere Indian Ocean based on Lebreton et al. (2017). (b) Total plastic waste input in the northern hemisphere Indian Ocean for each month. Lebreton et al. (2017) based the monthly variation of plastic input on seasonal variations in river discharges.





### 2.2.1 Long-term simulation

We run 21 year particle tracking simulations to determine the dynamics of plastics released in the NIO. During the first year of the simulation, we release particles into the NIO from river plastic source locations (Figure 2a; Lebreton et al., 2017). Several of the source locations available from Lebreton et al. (2017) are located on land grid cells in HYCOM. We prevent releasing particles on or very close to land by increasing the HYCOM land mask with one grid cell and then moving any release locations on land to the nearest ocean grid cell (Figure A2). We include the monthly variation of plastic waste input from rivers (Figure

2b) in our simulation by releasing particles on the first day of every month. A single particle in our simulation represents 1 tonne of plastic waste. After inputting particles for the first year, we then run the simulation for an additional 20 years to determine the influence of the Indian Ocean dynamics on particle transport.

We release simulated particles in 1995, because HYCOM data is available from then onwards and we want to run simulations for as long as possible using this dataset. This does not necessarily mean that the plastic waste input estimated by Lebreton

et al. (2017) is representative for 1995. We are interested in the large-scale and long-term dynamics of plastics in the NIO rather than in the behaviour of plastics during a specific time period, so this is not an issue for this paper.

### 2.2.2 Monsoonal simulation

In addition to long-term dynamics, we are also interested in the influence of the monsoon system on the transport of plastics. One of the dominant climate modes that influences atmospheric and oceanic dynamics in the NIO is the Indian Ocean Dipole

(IOD; Saji et al., 1999; Ashok and Guan, 2004; Schott et al., 2009). To determine the influence of the monsoon season on plastic transport in the NIO, we run an additional simulation during neutral IOD conditions. Both 2008 and 2009 were neutral IOD years, with relatively low values of the Dipole Mode Index (DMI; Figure A3; Saji et al., 1999). We therefore release particles in 2008 and continue the simulation to the end of 2009. We use the simulation results of the second simulation year to illustrate the influence of the monsoon system on plastic transport in the NIO (section 3.1).

### 145 2.3 Beaching

We do not implement any specific beaching behaviour during the particle tracking simulation. Instead, particles remain adrift in the simulation and we apply beaching conditions to each particle afterwards, using 5 day outputs of particle locations. This way, we can easily implement different beaching conditions and determine the sensitivity of our results without running a large number of simulations.

Beaching of plastics is highly complex and strongly influenced by small-scale coastal ocean dynamics (Isobe et al., 2014), as well as the local morphology of the coastline (Zhang, 2017). In addition, beached plastics do not necessarily remain beached but can return to the ocean (Zhang, 2017; Lebreton et al., 2019). Plastics also fragment relatively easily while exposed to sunlight and high temperatures on beaches (Andrady, 2011), as well as breaking waves near coastlines (Zhang, 2017). As a result of changes in the material characteristics (shape, size, density) of plastics, their response to ocean dynamics may also

change (e.g. Maximenko et al., 2012; van der Mheen et al., 2019). It is beyond the purpose and scope of this paper to account



for these complex and small-scale beaching dynamics of plastics. Instead, our goal is to provide indicative large-scale spatial patterns of beaching plastics in the NIO.

We define that particles within a distance $\Delta x$ of any coastline, and moving towards the coastline (defined as a decreasing distance to the coast), beach randomly with a specific probability $p$. The beaching probability can vary between 0 (no particles

beach) and 1 (all particles within a distance $\Delta x$ of a coastline beach) per 5 days. If a particle beaches, it remains beached and its location is fixed for the remainder of the simulation. Similar methods to account for beaching plastics in large-scale simulations have been used in other studies (Lebreton et al., 2019).

We use the distance to the nearest coastline from GSHHG-v2.3.7 data (Figure A4; Wessel and Smith, 1996) to determine the distance of particles to a coastline. This dataset has a horizontal resolution of 1 arcminute. The high resolution allows us to

include the coastlines of small islands in our beaching analysis.

### 2.3.1   Sensitivity to beaching distance $\Delta x$ and probability $p$

We performed sensitivity analyses of our results for different values of both the beaching distance $\Delta x$ and probability $p$. We used $\Delta x = [2, 4, 8, 16]$ km with $p = 0.50/5$ days to determine the sensitivity of our results to beaching at different distances $\Delta x$ from the nearest coastline. Our results are not very sensitive to these different values of $\Delta x$ (Figure A5). We therefore use

a fixed value of $\Delta x = 8$ km (which is approximately the size of one HYCOM grid cell) for the rest of our analyses.

In contrast, our results are sensitive to different values of beaching probability $p$. We discuss this further in section 3.2 and present our results for different values of $p$.

## 3   Results

### 3.1   Monsoonal influence and escape mechanism from NIO to SIO

Particle tracking simulation results during neutral IOD conditions and without beaching illustrate the influence of the monsoon season on the transport of particles in the NIO. During the northeast monsoon season, particles are transported from the Bay of Bengal towards the Arabian Sea by the Northeast Monsoon Current (NMC, Figure 3a). Particles are present throughout both the Arabian Sea and the Bay of Bengal during the following intermonsoon period (Figure 3b). During the southwest monsoon season, particles are transported from the Arabian Sea towards the Bay of Bengal by the Southwest Monsoon Current (SMC,

Figure 3c). Most particles are in the Bay of Bengal during this season, and remain there during the next intermonsoon period as eastward Wyrtki jets (WJ) develop around the equator (Figure 3d).

These simulation results indicate that particles leave the Arabian Sea depending on the monsoon season. In contrast, there are relatively high particle concentrations in the Bay of Bengal throughout the year. Although there is no region of consistent downwelling in the Bay of Bengal (and therefore no persistent accumulation of plastics), anti-cyclonic and cyclonic gyres

develop in the bay throughout the year (Paul et al., 2009), which may trap plastics. In addition, the annual mean flow along the equator is eastwards, directed from the Arabian Sea towards the Bay of Bengal (Schott et al., 2009; de Vos et al., 2014).


These simulation results also indicate an "escape" mechanism for particles to cross the equator from the NIO into the SIO. Particles mainly cross the equator during the intermonsoon period following the southwest monsoon season (Figure 3d). During this period, the WJ are at their strongest (Qui and Yu, 2009) and particles are transported eastwards along the equator. Particles

cross the equator with the south-eastward flowing South Java Current (SJC) and connect with the westward flowing South Equatorial Current (SEC).

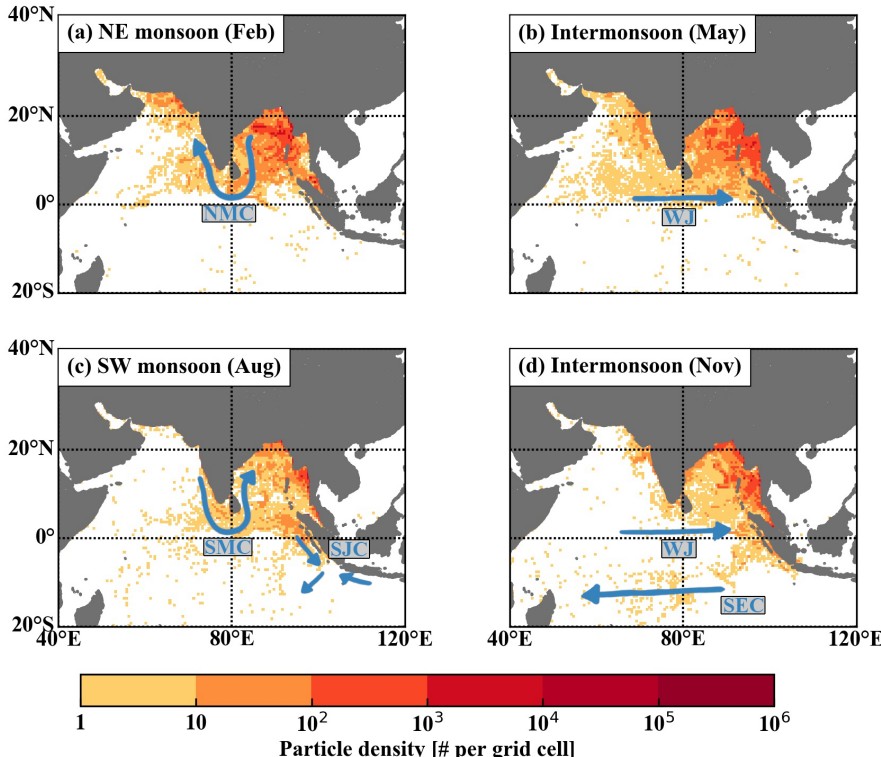

**Figure 3.** Particle density of simulated particles released from river source locations in the northern hemisphere Indian Ocean during neutral Indian Ocean Dipole conditions and without beaching at the end of: (a) the northeast monsoon season (February); (b) the intermonsoon period transitioning from the northeast to the southwest monsoon (May); (c) the southwest monsoon season (August); and (d) the intermonsoon period transitioning from the southwest to the northeast monsoon (November). Blue arrows indicate relevant ocean surface currents labeled with their abbreviations: Northeast Monsoon Current (NMC); Wyrtki jets (WJ); Southwest Monsoon Current (SMC); South Java Current (SJC); South Equatorial Current (SEC).





## 3.2 Beaching

As described in section 2.3, we allow simulated particles to randomly beach with a probability $p$ if they are moving towards the coast within a distance $\Delta x = 8$ km of a coastline. Realistic beaching probabilities of plastics are unknown and are beyond the scope of this paper to determine. We therefore consider particle tracking simulation results for a beaching probability of $p = 0.50/5$ days, as well as a "high" beaching probability of $p = 0.95/5$ days, and a "low" beaching probability of $p = 0.05/5$ days.

In both the simulation with high beaching probability and with a beaching probability of $p = 0.50/5$ days, almost all particles beach in the NIO within 3 years (Figure 4a and 4b). Only approximately 0.6 % of all particles cross from the NIO into the SIO in the high beaching probability simulation, compared to about 1 % of all particles in the simulation with beaching probability of $p = 0.50/5$ days. In the simulation with low beaching probability, around 86 % of all particles beach in the NIO after approximately 10 years (Figure 4c). About 5.7 % of all particles cross the equator into the SIO in this simulation, where they either beach (4.2 %) or end up in the subtropical SIO garbage patch (1.5 %).

### 3.2.1 Countries most affected

Which countries are most heavily affected by beaching particles released from the NIO depends on the beaching probability $p$. Nevertheless, there are some noteworthy general results and trends. Countries bordering the Bay of Bengal are consistently and heavily affected both for high and low beaching probability (Figures 5a and 5c, respectively). For high beaching probability, this is most likely due to the large source locations of particles in the Bay of Bengal (Figure 2a). For low beaching probability however, this is more likely a result of ocean dynamics in the region. As shown in section 3.1 (Figure 3), there are particles in the Bay of Bengal throughout the year, which are therefore likely to beach in the region.

Connectivity matrices (such as used by e.g. Escalle et al., 2019) showing the percentage of beached particles originating from different countries confirm this. For high beaching probability, particles that beach in specific countries mainly originate from that same country (Figure 5e, high percentages along the diagonal). In contrast, for low beaching probability, beached particles originate from multiple different countries (Figure 5g). In the Bay of Bengal, notable exceptions to this are Bangladesh and Malaysia, for which > 90 % of beached plastics originate from their own country, even for low beaching probability.

The countries that are among the top 15 that receive the most beached particles for all beaching probability $p$ values are: Bangladesh, Myanmar, India, Malaysia, Indonesia, Sri Lanka, Thailand, Pakistan, the Maldives, and Somalia (Table A2). Of these, only Somalia does not border the Bay of Bengal and does not have significant nearby inputs of plastic waste from rivers (Figure 2a). For low beaching probabilities, most particles beaching in Somalia originate from countries bordering the Bay of Bengal (Figure 5g). These particles most likely end up near Somalia as they are transported westward by the North Equatorial Current and the Somali Current during the northeast monsoon season.

The Maldives is also noteworthy, as it receives a relatively large percentage of particles for almost all values of $p$, even though it has no river plastic sources of its own. Because both the Northeast Monsoon Current (NMC) and the Southwest Monsoon Current (SMC) flow past the Maldives in reversing directions, it is not unexpected that the Maldives is heavily affected by


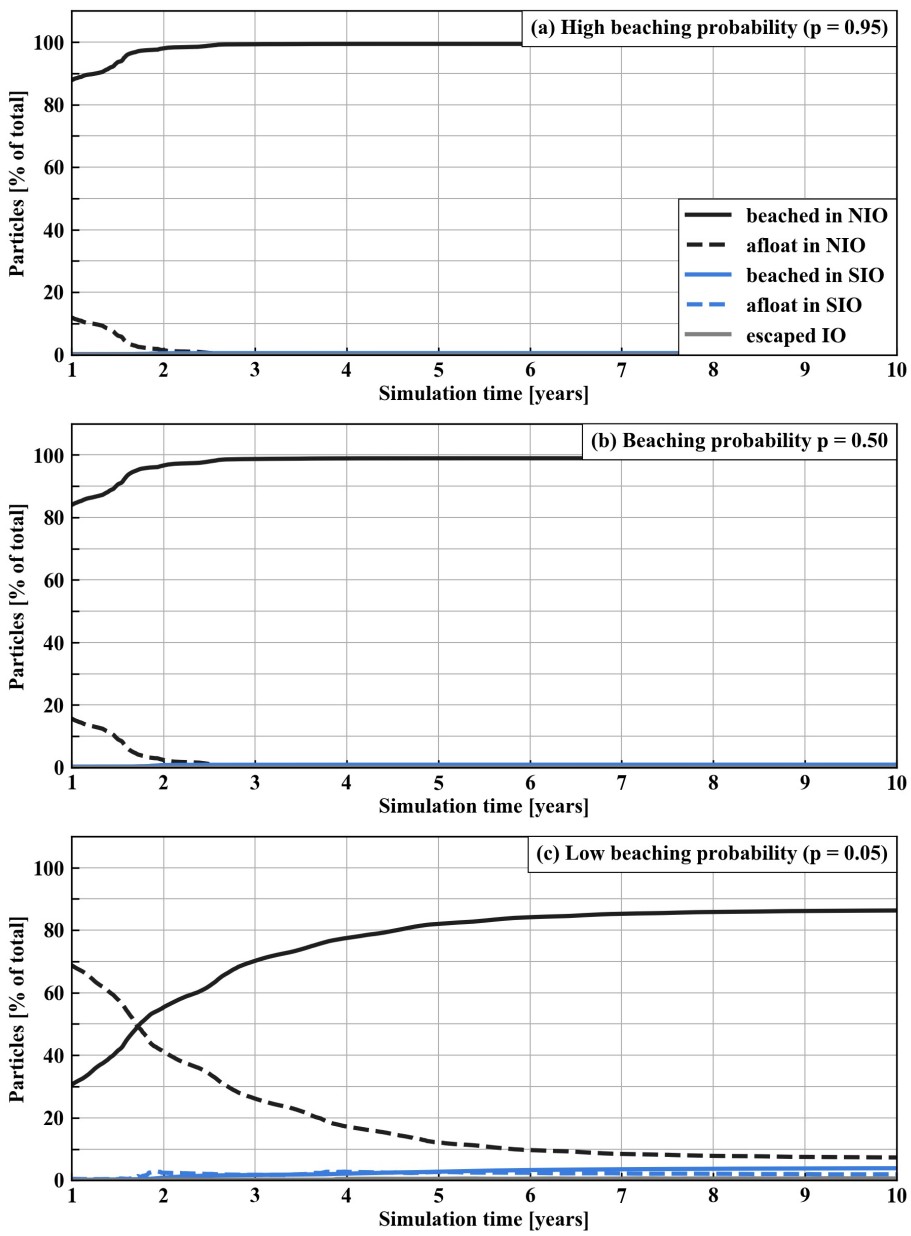

**Figure 4.** Percentage of total simulated particles as a function of the simulation duration that have: beached in the northern (NIO) or southern hemisphere Indian Ocean (SIO); are afloat in the NIO or SIO; or that have left the Indian Ocean entirely, for: (a) a high beaching probability of $p = 0.95/5$ days; (b) a beaching probability of $p = 0.50/5$ days; (c) a low beaching probability of $p = 0.05/5$ days. Percentages are shown after all particles have been released after 1 year of simulation, and up to 10 years of simulation, after which the simulation results have reached a steady state in all cases.





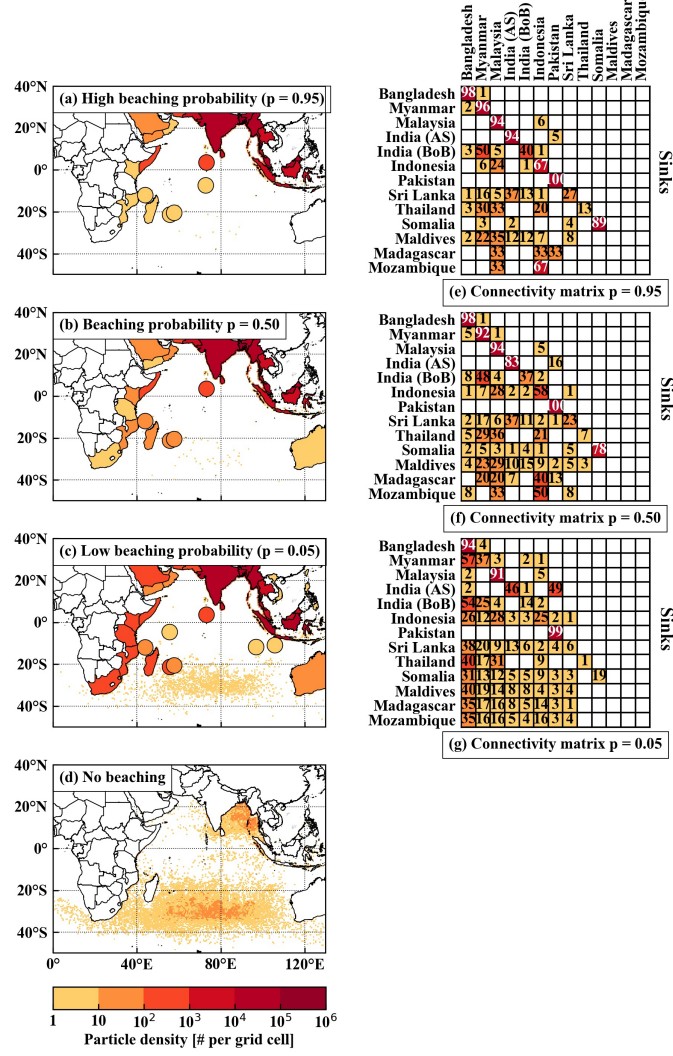

**Figure 5.** Density of beached particles per country or island and density of particles in the ocean per $0.5 \times 0.5°$ grid cell for particles released from river source locations in the northern hemisphere Indian Ocean after 21 years of simulation, with: (a) high beaching probability, $p = 0.95/5$ days (10 year animation of simulation results available at http://doi.org/10.5446/47058); (b) beaching probability of $p = 0.50/5$ days (10 year animation of simulation results available at http://doi.org/10.5446/47057); (c) low beaching probability, $p = 0.05/5$ days (10 year animation of simulation results available at http://doi.org/10.5446/47056); (d) no beaching. Filled circles highlight islands which do not clearly show up on the map otherwise, from north to south these represent: Maldives, Seychelles, British Indian Ocean Territory, Christmas Island, Cocos (Keeling) Islands, Comoros, Mauritius, and Réunion. Connectivity matrices showing the percentage of particles that beach in selected countries (rows) against countries of origin (columns), for: (e) high beaching probability, $p = 0.95/5$ days; (f) beaching probability of $p = 0.50/5$ days; and (g) low beaching probability, $p = 0.05/5$ days. In these matrices, India is split into a western (bordering the Arabian Sea, "India (AS)") and an eastern side (bordering the Bay of Bengal, "India (BoB)"). Because not all countries with river plastic sources are shown, percentages are rounded to integer numbers, and percentages below 1 % are omitted, the sum of each row does not always precisely equal 100 %.





beaching particles. Similarly, Sri Lanka is also affected by beaching particles from multiple source countries as the NMC and SMC flow past.

For decreasing beaching probabilities $p$, a larger percentage of particles crosses from the NIO into the SIO and several countries and islands in the SIO are increasingly affected by beaching particles (Table A2). Most notable among these are Madagascar and Mozambique, which are among the top 15 most affected countries for beaching probabilities $p \leq 0.225/5$
days.

## 4 Discussion

The aim of this paper is to determine what happens to plastics entering the NIO from rivers and which countries and islands are most heavily affected by beaching plastics. Our particle tracking simulation results illustrate that particles move between the Arabian Sea and the Bay of Bengal depending on the monsoon season. During the northeast monsoon season large amounts
of particles are present in the Arabian Sea as they are transported from the Bay of Bengal by the Northeast Monsoon Current (NMC). In contrast, during the southwest monsoon season particles are largely depleted from the Arabian Sea by the Southwest Monsoon Current (SMC) and move into the Bay of Bengal. Despite the annual back and forth movement, particles remain present year-round in the Bay of Bengal. This is possibly a result of the annual mean eastward flow in the equatorial region (Schott et al., 2009) as well as anti-cyclonic and cyclonic gyres that develop in the Bay of Bengal throughout the year (Paul
et al., 2009), which may trap plastics.

Countries bordering the Bay of Bengal are consistently and heavily affected by beaching plastics. Specifically, Bangladesh, Myanmar, India, Malaysia, Indonesia, Sri Lanka, Thailand, Pakistan, the Maldives, and Somalia are in the top 15 most affected countries in all our simulations. For high beaching probabilities, all particles beach in the NIO within 3 years. In this case, the locations where particles beach is mainly a result of large plastic sources in the region, and plastics mainly beach in their
country of origin. However, for low beaching probabilities, this is more likely a result of ocean dynamics, and beached plastics originate from multiple different countries. Because the NIO dynamics concentrate plastics in the Bay of Bengal, bordering countries are affected by beaching even on long timescales of $\mathcal{O}(10)$ years.

Somalia and the Maldives are specifically noteworthy countries affected by beaching plastics from the NIO. Somalia does not border the Bay of Bengal and does not have any large nearby sources of plastic. Nevertheless, large amounts of particles
consistently beach here. For low beaching probabilities, beached river plastics in Somalia mainly originate from countries that border the Bay of Bengal. The westward flowing North Equatorial Current and the south-westward flowing Somalia Current likely transport plastics to Somalia during the northeast monsoon season. The Maldives is noteworthy because the NMC and the SMC transport particles back and forth past the islands twice a year, which increases the likelihood of plastics beaching here. The same is true for Sri Lanka.

For low beaching probabilities, up to 5% of particles "escape" from the NIO into the SIO. This mainly occurs on the eastern side of the NIO basin during the intermonsoon period following the southwest monsoon season (September, October, November). We propose the following mechanism for particles crossing from the NIO into the SIO: (1) particles are trans-





ported eastwards by equatorial Wyrtki jets during the intermonsoon period; (2) particles are transported south-eastwards across the equator by the South Java Current (SJC); (3) particles are transported south-westwards as the SJC feeds into the South

Equatorial Current (SEC); and (4) particles are transported westwards by the SEC into the subtropical SIO.

Particles that cross from the NIO into the SIO mainly beach on eastern African coastlines or accumulate in the subtropical SIO garbage patch. Madagascar and Mozambique are most notably increasingly affected as more particles cross into the SIO.

Countries and islands in the SIO will of course also be affected by beaching plastics entering the ocean from source locations in the SIO (Figure 6a). In this case, the most affected countries in the SIO are similar to those affected by plastics escaping

from the NIO into the SIO (Figure 6b, 6c, and 6d). Notable exceptions to this are the Cocos (Keeling) Islands and Christmas Island, both of which are more severely affected by beaching particles originating from the SIO (especially with high beaching probability, Figure 6b). Connectivity matrices indicate that particles mostly beach in their country of origin, or come from Indonesia (Figure 6e, 6f, and 6g). Besides beaching in the SIO, plastics entering the SIO also accumulate in the subtropical garbage patch (up to 5 % for high beaching probability versus 36 % for low beaching probability). Particles can also cross

the equator and beach in NIO countries, although this occurs less frequently than plastics crossing from the NIO into the SIO (around 2 % crossing from the SIO into the NIO, compared to up to 5 % crossing from the NIO into the SIO for low beaching probabilities). Finally, particles entering the SIO also escape the Indian Ocean entirely: up to 2 % for high beaching probability and up to 7 % for low beaching probability.

Our results indicate that a large percentage of plastics end up on coastlines in the Indian Ocean. In our simulations with a

high beaching probability, 100% of particles beach in the NIO within 3 years. Up to 90% of particles beach in either the NIO or SIO within 10 years in our simulations with a low beaching probability. These results are in good general agreement with those of Lebreton et al. (2019), who showed that roughly 67% of all global plastic waste ended up on coastlines. Lebreton et al. (2019) therefore suggested that the large mismatch between the estimated amount of plastic entering the ocean globally and the total estimated amount of plastic floating on the ocean surface (the "missing plastic", van Sebille et al., 2015), can be

explained by plastics stored on coastlines. However, our simulations illustrate that results are sensitive to different beaching conditions, specifically the beaching probability. To determine if beached plastics can indeed explain the whereabouts of the missing plastic, it is therefore important to apply reliable beaching conditions.

The importance of coastal dynamics in the transport of plastics to the open ocean was recently demonstrated by Zhang et al. (2020), who found that as a result of tidal dynamics only roughly 20% of simulated particles released around the East China

Sea were transported to the open ocean. Pawlowicz et al. (2019) showed that ocean surface drifters in an estuary ran aground on timescales much shorter than the transport time to the open ocean. Both of these studies illustrate the importance of local dynamics in transporting plastics to the ocean. A better understanding of the overall effect of these dynamics as well as a method to apply them on large scales (for example using a realistic beaching probability) is therefore needed to improve global and basin-scale models of beaching plastics.

van der Mheen et al. (2019) showed that different transport mechanisms, due to wind and waves, have a significant influence on the accumulation of buoyant debris in the subtropical SIO garbage patch. In this paper, we only considered the effect of ocean surface currents on the transport of river plastics entering the NIO. It is not straightforward to apply the same beaching





methodology when simulations are forced not only by ocean surface currents, but by wind and wave effects as well. This is because, in contrast to ocean surface currents, the transport due to wind and Stokes drift can be directed perpendicular to

95 coastlines. This means that including wind or wave effects adds a physical mechanism to the beaching of particles. However, in our methodology we assume that there are no physical beaching processes in the particle tracking simulations, and beaching is included purely as a specified probability acting a certain distance from the coastline. This assumption is reasonable when particles are forced only by ocean surface currents, but it is no longer valid when wind or Stokes drift forcing is included as well. The best method to include wind and wave effects in these beaching simulations needs more careful consideration and

300 extended analysis, which we will do in future work.

However, because both wind and ocean surface currents in the NIO are driven by the monsoon system, we do not expect the influence of including either windage or Stokes drift to have such a significant effect as in the SIO. For example, although the timescales on which beaching occurs will likely change by including windage or Stokes drift, the main dynamics of particles moving between the Arabian Sea and the Bay of Bengal depending on the monsoon season should remain the same.

305 Finally, measurements of plastics on coastlines are needed to confirm and improve numerical modeling results. Although multiple studies sampled plastics on beaches throughout the Indian Ocean (Ryan, 1987; Slip and Burton, 1991; Madzena and Lasiak, 1997; Uneputty and Evans, 1997; Barnes, 2004; Jayasiri et al., 2013; Duhec et al., 2015; Nel and Froneman, 2015; Bouwman et al., 2016; Kumar et al., 2016; Imhof et al., 2017; Lavers et al., 2019), the different sampling methods and timescales mean that their results are difficult to compare quantitatively. In addition, standing stock measurements are of

310 limited use because they provide no information about the time period over which plastics may have accumulated on beaches. Ideally, long-term measurements during different conditions and along different types of coastline are needed.



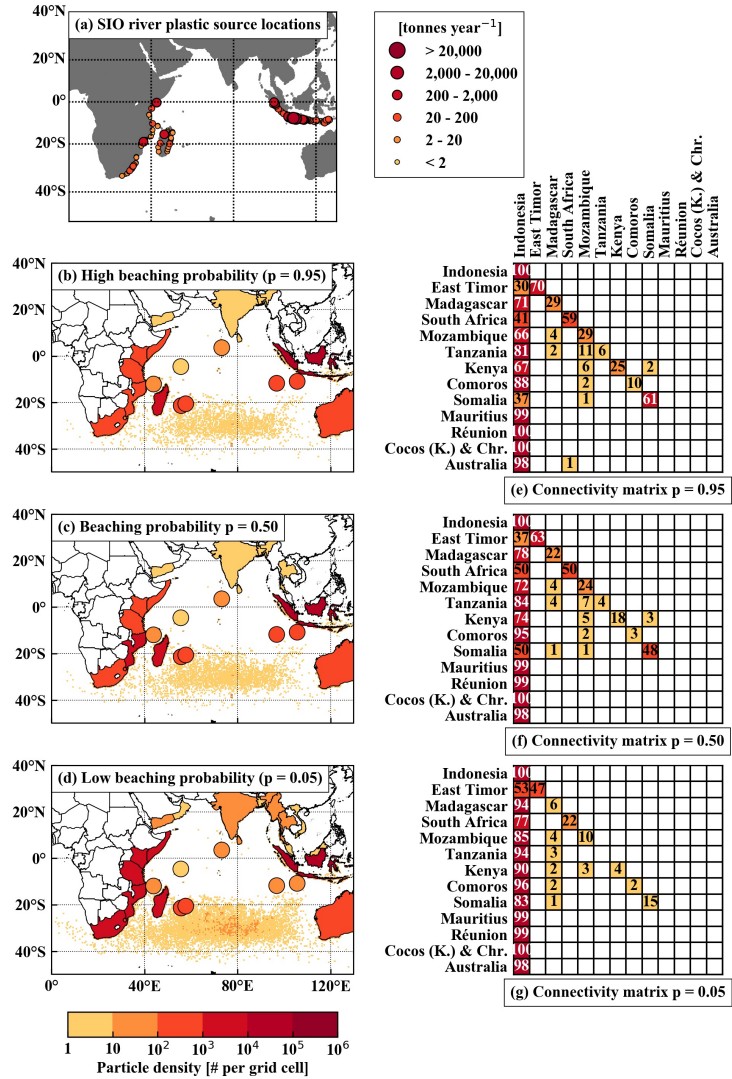

**Figure 6.** (a) Locations of plastic waste input from rivers in the southern hemisphere Indian Ocean based on Lebreton et al. (2017). Density of beached particles per country or island and density of particles in the ocean per $0.5 \times 0.5°$ grid cell for particles released from river source locations in the southern hemisphere Indian Ocean after 21 years of simulation, with: (b) high beaching probability, $p = 0.95/5$ days (10 year animation of simulation results available at: http://doi.org/10.5446/47058); (c) beaching probability of $p = 0.50/5$ days (10 year animation of simulation results available at: http://doi.org/10.5446/47057); and (d) low beaching probability, $p = 0.05/5$ days (10 year animation of simulation results available at http://doi.org/10.5446/47056). Filled circles highlight islands which do not clearly show up on the map otherwise, from north to south these represent: Maldives, Seychelles, British Indian Ocean Territory, Christmas Island, Cocos (Keeling) Islands, Comoros, Mauritius, and Réunion. Connectivity matrices showing the percentage of particles that beach in selected countries (rows) against countries of origin (columns), for: (e) high beaching probability, $p = 0.95/5$ days; (f) beaching probability of $p = 0.50/5$ days; and (g) low beaching probability, $p = 0.05/5$ days. Because not all countries with river plastic sources are shown, percentages are rounded to integer numbers, and percentages below 1 % are omitted, the sum of each row does not always precisely equal 100 %.





## 5   Conclusions

The aim of this paper is to determine what happens to plastics that enter the NIO from rivers. Our particle tracking simulation results show that plastics move back and forth between the Bay of Bengal and the Arabian Sea depending on the monsoon season. During the southwest monsoon season, the Arabian Sea almost completely depletes of particles as they are transported to the Bay of Bengal by the Southwest Monsoon Current. In contrast, there are relatively high concentrations of particles present in the Bay of Bengal year round. This may be due to the annual mean eastward flow in the equatorial region (Schott et al., 2009) as well as anti-cyclonic and cyclonic gyres in the Bay of Bengal (Paul et al., 2009) trapping plastics.

Particles move close to coastlines as they move between the Arabian Sea and the Bay of Bengal. When we allow simulated particles to beach with a "high" beaching probability ($p = 0.95/5$ days), all particles beach in the NIO within 3 years, mostly in their country of origin. For "low" beaching probability ($p = 0.05/5$ days), 86 % of particles beach in the NIO in 10 years. In most countries, beached river plastics originate from multiple different countries for low beaching probability. Countries bordering the Bay of Bengal are heavily affected by beaching particles, likely because ocean dynamics concentrate particles in this region. Somalia and the Maldives are also consistently affected by beaching particles, even though they have no or little river sources of plastics of their own. In the case of the Maldives, this is a result of the Southwest Monsoon Current and the Northeast Monsoon Current transporting particles back and forth past the islands twice a year. In the case of Somalia, the North Equatorial Current and the Somalia Current likely transport particles originating from countries in the Bay of Bengal towards the Somalian coast.

In simulations with low beaching probability, up to 5 % of particles "escape" from the NIO into the SIO, where they predominantly beach along eastern African coastlines. Particles mostly pass the equator along the eastern side of the Indian Ocean basin during the intermonsoon period following the southwest monsoon season (September, October, November). We suggest the following mechanism for their escape from the NIO into the SIO: (1) particles are transported eastwards by equatorial Wyrtki jets; (2) particles are transported south-eastwards across the equator by the South Java Current; (3) particles are transported south-westwards where the South Java Current feeds into the South Equatorial Current; and (4) particles are transported westwards into the subtropical SIO by the South Equatorial Current.

*Code and data availability.*  Ocean surface currents from the HYCOM+NCODA Global $1/12°$ Reanalysis dataset are available from: www.hycom.org/data/glbv0pt08/expt-53ptx. Distances to the nearest coastline based on the GSHHS dataset are available from: www.soest.hawaii.edu/pwessel/gshhg/. We obtained values of the Indian Ocean Dipole Mode Index from: stateoftheocean.osmc.noaa.gov/sur/ind/dmi.php. Our code to run particle tracking simulations with OceanParcels and to apply beaching conditions is available under an MIT license: www.github.com/mheen/io_beaching.





*Video supplement.* Animations of 10 year particle tracking simulation results with particles entering the Indian Ocean from river plastic sources are available with beaching occurring at a distance $\Delta x = 8$km to the nearest coastline with a probability of: $p = 0.05/5$days (http://doi.org/10.5446/47056); $p = 0.50/5$days (http://doi.org/10.5446/47057); and $p = 0.95/5$days (http://doi.org/10.5446/47058).



## Appendix A: Additional figures and tables

This appendix provides:

1.   Table A1: Overview of studies that sampled plastics on beaches in the Indian Ocean. This table illustrates that a quantitative comparison between studies is difficult because of different methods and timescales of sampling.

2.   Figure A1: Boundaries of the northern and southern hemisphere Indian Ocean used in analyses discussed in the main article.

3.   Figure A2: Example of the method used to move original source locations of plastic waste a suitable distance away from land for release of particles in the particle tracking simulations.

4.   Figure A3: Indian Ocean Dipole Mode Index used to determine neutral Indian Ocean Dipole years to run particle tracking simulations to determine the influence of different monsoon seasons on particle transport.

5.   Figure A4: Distance to the nearest coastline used to determine beaching of particles.

6.   Figure A5: Sensitivity analysis results for beaching at different distances to the coast $\Delta x$. Results are not very sensitive to different values of $\Delta x$, so we use $\Delta x = 8$ km for analyses in the main article.

7.   Table A2: Top 15 most affected countries by beaching particles for beaching with different probabilities $p$.




**Table A1.** Brief overview of studies that sampled plastics on beaches in the Indian Ocean, including methods and findings. Different studies use many different methods and units, and sampling was done on very different timescales.

| Location | Plastic items | units | Transect size [m] | Sampling time | Standing stock or cleared | Size ranges | Excavation | Beach characteristics | Reference |
|---|---|---|---|---|---|---|---|---|---|
| Transkei, South Africa | 1.2 - 8.1 | $\#\,m^{-1}\,year^{-1}$ | 3 x 8-30 | April 1994-1995 monthly | cleared | - | raked | undeveloped | Madzena and Lasiak (1997)[1] |
| Prince Edward Island | 0.19 | $\#\,m^{-1}\,year^{-1}$ | - | - | - | - | - | - | Ryan (1987)[1] |
| Marion Island | 0.055 | $\#\,m^{-1}\,year^{-1}$ | - | - | - | - | - | - | Ryan (1987)[1] |
| Heard Island | 0.015 | $\#\,m^{-1}\,year^{-1}$ | - | - | - | - | - | - | Slip and Burton (1991)[1] |
| Macquarie Island | 0.1 | $\#\,m^{-1}\,year^{-1}$ | - | - | - | - | - | - | Slip and Burton (1991)[1] |
| Jakarta Bay, Indonesia | 90 | $\#\,m^{-1}\,year^{-1}$ | - | - | - | - | - | - | Uneputy and Evans (1997)[1] |
| Negombo, Sri Lanka | 1.55 | $\#\,m^{-1}\,year^{-1}$ | 200 | 1996-2002 yearly | cleared | >1 cm2 | part buried | windward low/no population | Barnes (2004) |
| Ari Atoll, Maldives | 1.12 | $\#\,m^{-1}\,year^{-1}$ | 200 | 1996-2002 yearly | cleared | >1 $cm^2$ | part buried | windward low/no population | Barnes (2004) |
| Pemba Island, Tanzania | 1.89 | $\#\,m^{-1}\,year^{-1}$ | 200 | 1996-2002 yearly | cleared | >1 $cm^2$ | part buried | windward low/no population | Barnes (2004) |
| Diego Garcia | 0.89 | $\#\,m^{-1}\,year^{-1}$ | 200 | 1996-2002 yearly | cleared | >1 $cm^2$ | part buried | windward low/no population | Barnes (2004) |
| Christmas Island | 21 | $\#\,m^{-1}\,year^{-1}$ | 200 | 1996-2002 yearly | cleared | >1 $cm^2$ | part buried | windward low/no population | Barnes (2004) |
| Cocos (Keeling Islands) | 6.01 | $\#\,m^{-1}\,year^{-1}$ | 200 | 1996-2002 yearly | cleared | >1 $cm^2$ | part buried | windward low/no population | Barnes (2004) |
| Quirimba Island, Mozambique | 1.34 | $\#\,m^{-1}\,year^{-1}$ | 200 | 1996-2002 yearly | cleared | >1 $cm^2$ | part buried | windward low/no population | Barnes (2004) |
| Rodrigues Island | 4.41 | $\#\,m^{-1}\,year^{-1}$ | 200 | 1996-2002 yearly | cleared | >1 $cm^2$ | part buried | windward low/no population | Barnes (2004) |
| Nosy Ve, Madagascar | 0.60 | $\#\,m^{-1}\,year^{-1}$ | 200 | 1996-2002 yearly | cleared | >1 $cm^2$ | part buried | windward low/no population | Barnes (2004) |
| Inhaca Island, Mozambique | 0.60 | $\#\,m^{-1}\,year^{-1}$ | 200 | 1996-2002 yearly | cleared | >1 $cm^2$ | part buried | windward low/no population | Barnes (2004) |
| Mumbai, India | 68.8 | $\#\,m^{-2}\,week^{-1}$ | 0.5 x 0.5 | May 2011-March 2012 bimonthly | - | 1-5 mm 5-20 mm 21-100 mm >100 mm | top 2 cm | highly populated | Jayasiri et al. (2013) |
| Alphonse Atoll, Seychelles | 4.7 | $\#\,m^{-1}\,week^{-1}$ | 500 | 21 June-2 August 2013 weekly | cleared | - | - | windward low/no population | Dubuc et al. (2015) |
| South-east coast South Africa | 689-3308 | $\#\,m^{-2}$ | - | - | - | 80 $\mu m$-5 mm | top 5 cm | 12 beaches in bays 9 beaches on open coast | Nel and Froneman (2015) |
| St. Brandon's Rock, Mauritius | 0.76 | $\#\,m^{-1}$ | varying | October 2010 and 2014 | standing (?) | >5 mm | - | low/no population | Bouwman et al. (2016) |
| Chennai, India | 1.37 | $\#\,m^{-2}\,2-weeks^{-1}$ | 100 | March and April 2015 bimonthly | cleared | - | part buried | highly populated | Kumar et al. (2016) |
| Vavvaru Island, Maldives | 35.8 | $\#\,m^{-2}\,day^{-1}$ | 1 x 1 | 21-27 June 2015 daily | cleared and standing | 1-5 mm 5-25 mm >25 mm | top 1 cm | low/no population open coast | Imhof et al. (2017) |
| Cocos (Keeling Islands) | 4.72-2506 | $\#\,m^{-2}$ | 30 x 6 | 13-24 March 2017 21 September 2017 single sampling | standing | - | top 10 cm | low/no population | Lavers et al. (2019) |

[1] Contained in Barnes (2004)



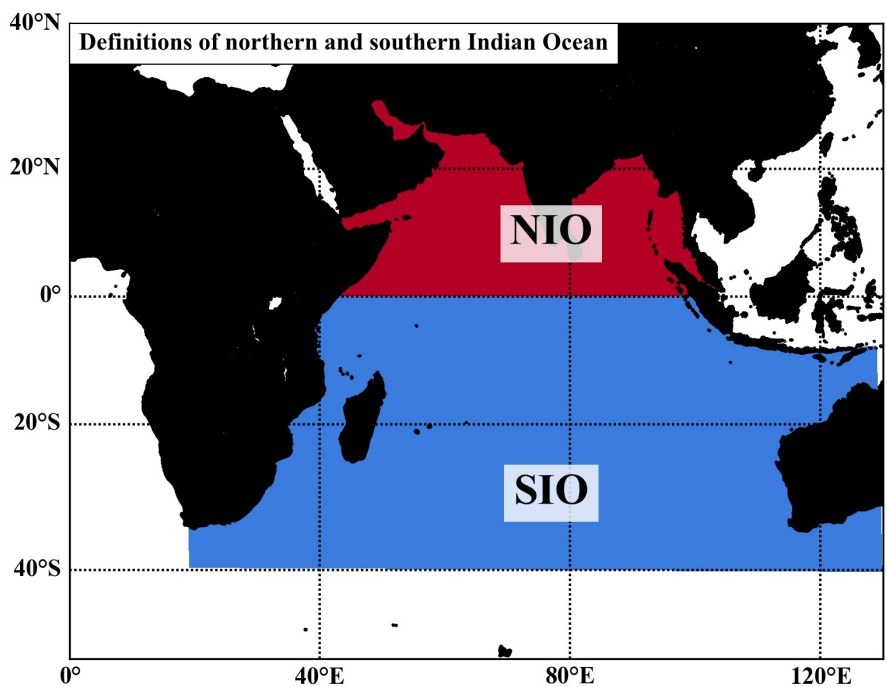

**Figure A1.** Definition of the northern hemisphere Indian Ocean (NIO) and southern hemisphere Indian Ocean (SIO). We use these definitions to select release locations of particles from the NIO only and to determine the fate of particles during the simulation (e.g. beached or floating in the NIO or SIO).

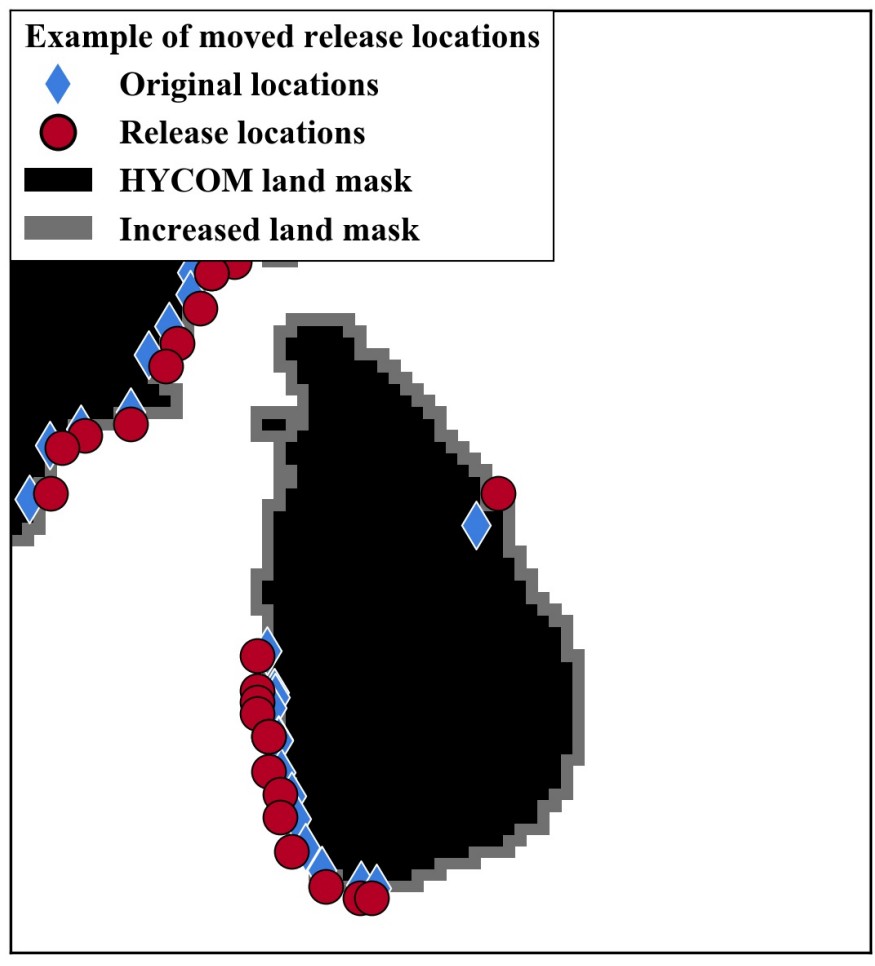

**Figure A2.** Example of original river source locations estimated by Lebreton et al. (2017) and moved release locations in relation to the HYCOM land mask around Sri Lanka. Release locations are shifted compared to original source locations where necessary to prevent particles from being released on or too close to land in particle tracking simulations.





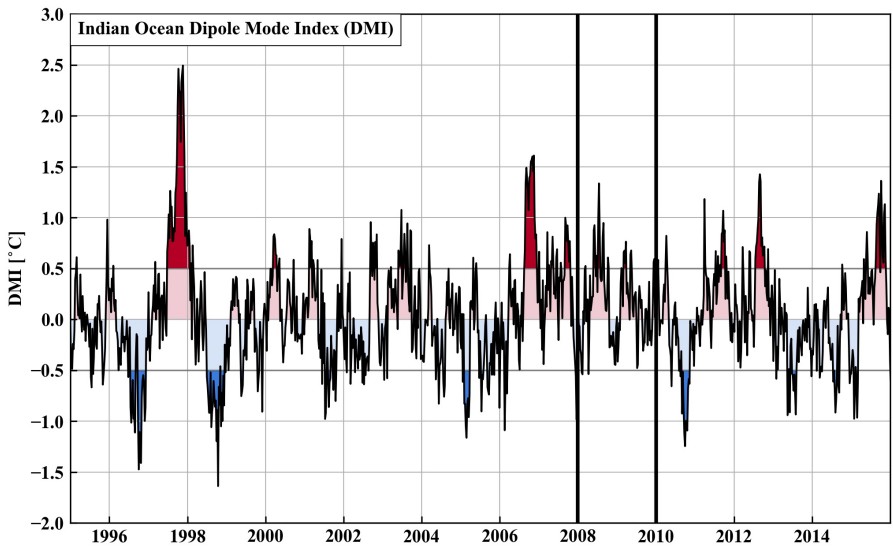

**Figure A3.** Indian Ocean Dipole Mode Index (DMI) as defined by Saji et al. (1999) and obtained from the National Oceanic and Atmospheric Administration. Red and blue shading indicate positive and negative modes of the Indian Ocean Dipole (IOD) respectively. We use 2008 and 2009 (marked between thick black vertical lines) as neutral IOD years to simulate the influence of monsoon seasons on the transport of plastics in the Indian Ocean.





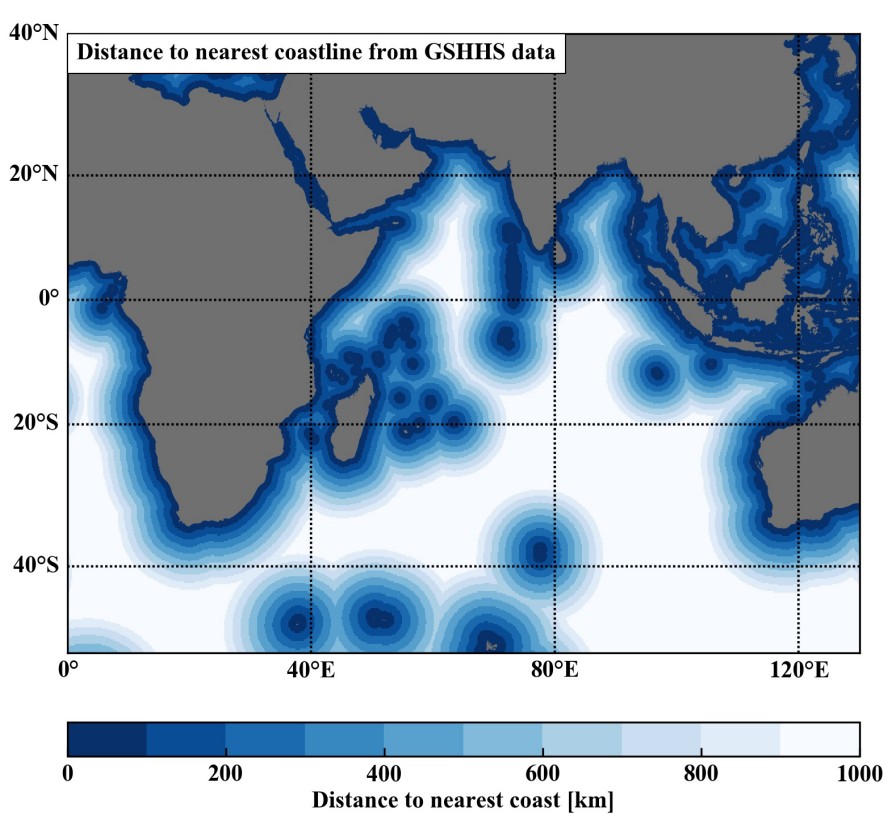

**Figure A4.** Distance to the nearest coastline based on GSHHG-v2.3.7 data (Wessel and Smith, 1996). We use this distance to determine beaching conditions for simulated particles.


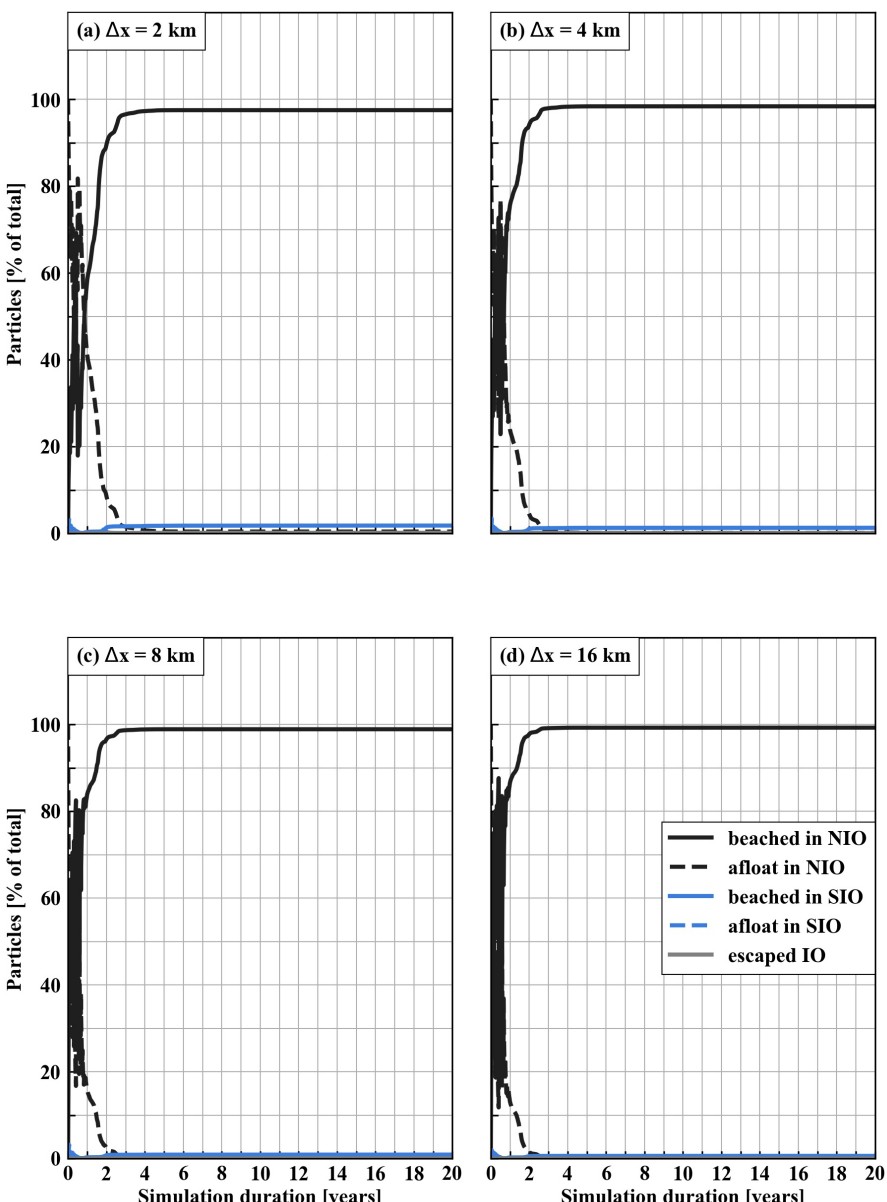

**Figure A5.** Sensitivity analysis results where beaching occurs with a probability $p = 0.50/5$ days for particles within a distance $\Delta x = [2, 4, 8, 16, 32]$ km to the nearest coastline that are moving towards the coast. Results are not very sensitive to different values for $\Delta x$, and we use $\Delta x = 8$ km as the default value in further simulations.





**Table A2.** Top 15 most affected countries by beaching plastics released from the northern hemisphere Indian Ocean from river sources. Results are shown for different beaching probabilities $p$.

| $p = 0.950/5$ days | | $p = 0.725/5$ days | | $p = 0.500/5$ days | | $p = 0.225/5$ days | | $p = 0.050/5$ days | |
|---|---|---|---|---|---|---|---|---|---|
| Country | Beached particles [% of total] | Country | Beached particles [% of total] | Country | Beached particles [% of total] | Country | Beached particles [% of total] | Country | Beached particles [% of total] |
| Bangladesh | 60 | Bangladesh | 60 | Bangladesh | 60 | Bangladesh | 56 | Myanmar | 30 |
| Myanmar | 13 | Myanmar | 14 | Myanmar | 14 | Myanmar | 15 | Bangladesh | 29 |
| India | 9.8 | India | 10 | India | 10 | India | 12 | India | 18 |
| Malaysia | 6.1 | Malaysia | 6.0 | Malaysia | 5.7 | Malaysia | 5.3 | Indonesia | 8.1 |
| Indonesia | 4.6 | Indonesia | 4.6 | Indonesia | 4.8 | Indonesia | 5.2 | Thailand | 3.4 |
| Sri Lanka | 1.2 | Sri Lanka | 1.3 | Thailand | 1.4 | Thailand | 2.1 | Malaysia | 3.0 |
| Pakistan | 1.2 | Pakistan | 1.2 | Sri Lanka | 1.4 | Sri Lanka | 1.5 | Sri Lanka | 2.2 |
| Thailand | 0.98 | Thailand | 1.1 | Pakistan | 1.1 | Pakistan | 0.91 | Madagascar | 0.36 |
| Maldives | 0.19 | Maldives | 0.19 | Maldives | 0.19 | Maldives | 0.18 | Pakistan | 0.34 |
| Kuwait | 0.14 | Kuwait | 0.14 | Kuwait | 0.13 | Kuwait | 0.12 | Somalia | 0.30 |
| Iran | <0.10 | Iran | <0.10 | Iran | <0.10 | Iran | 0.10 | Mozambique | 0.25 |
| Somalia | <0.10 | Somalia | <0.10 | Somalia | <0.10 | Somalia | <0.10 | Maldives | 0.19 |
| Saudi Arabia | <0.10 | Saudi Arabia | <0.10 | Saudi Arabia | <0.10 | Saudi Arabia | <0.10 | Kenya | 0.18 |
| Yemen | <0.10 | Yemen | <0.10 | Madagascar | <0.10 | Madagascar | <0.10 | Iran | 0.16 |
| Kenya | <0.10 | Kenya | <0.10 | Oman | <0.10 | Mozambique | <0.10 | Tanzania | 0.14 |



*Author contributions.* MvdM performed the research and prepared the manuscript. EvS and CP jointly supervised the work. All authors reviewed the manuscript.

*Competing interests.* The authors declare that they have no conflict of interest.

*Acknowledgements.* MvdM was supported by an Australian Government Research Training Program (RTP) Scholarship, a CFH & EA Jenkins Postgraduate Research Scholarship from the University of Western Australia, and an Ad Hoc Scholarship from the University of Western Australia. EvS was supported by the European Research Council (ERC) under the European Union's Horizon 2020 research and innovation programme (grant agreement No 715386).





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
