# Peer review of "Beaching patterns of plastic debris along the Indian Ocean rim"

_Ocean Science, 2020_

## Referee Comment (RC1) · Anonymous Referee #1 · 26 Jul 2020

Review of the manuscript "Beaching patterns of plastic debris along the Indian Ocean rim" by van der Mheen, van Sebille und Pattiaratchi

This manuscript is concerned with the fate of buoyant marine plastic debris entering the northern Indian Ocean from rivers due to advection by ocean currents and beaching. Due to the complicated and variable circulation patterns in the Indian Ocean which vary seasonally (monsoon) this is a complicated task.

The manuscript represents a substantial contribution to progress in the field and is very well suited for Ocean Science. Particle tracking in the Indian Ocean for such an extended period of time in the high-resolution model used gives insight into the pathways and connectivity of the whole region.

[Figure]

The scientific quality is excellent, the approach and methods are very appropriate and the results are clearly presented and very well discussed in the context of existing literature.

An eye-opener for every oceanographer interested in the surface circulation of the Indian Ocean are the animated simulations for the 3 different beaching probabilities for a period of 10 years.

Obviously, there are many open questions regarding the influence of wave-current interaction, Stokes drift, beaching processes, but they are all recognized and adequately addressed in the manuscript as open questions.

Typos: line 76 + 77: change Wrytki to Wyrtki

---

## Referee Comment (RC2) · Anonymous Referee #2 · 13 Aug 2020

The manuscript presents an interesting analysis of the transport and distribution of marine plastic debris from rivers into the northern Indian Ocean. The objective of the work is clear and the manuscript is well addressed and discussed. An interesting analysis is carried out to show how the results depend on the beaching methodology. The authors acknowledge that beaching of plastics is highly complex and that dynamics due to wind and waves are not considered in the simulations. These questions and their implications are identified and discussed in the manuscript. The manuscript represents a substantial contribution to scientific progress within the scope of Ocean Science and presents a high scientific and presentation quality.

However, I have some comments that I would like to be discussed in more detail: 1- As the authors mention, they only consider the effect of surface currents on the

transport of plastics in this study. This is correct, but this means that the results are representative of the marine plastic debris transported by surface currents. Buoyant items can be highly affected by wind, especially in coastal areas, where the wind can play an important role in the transport and beaching of marine debris. The authors would have to clarify the type of buoyant marine plastic debris under consideration and/or discuss in more detail how the results might change if the windage is included in the numerical simulations.

2- One of the main objectives of the study is to determine which countries and islands are most heavily affected by beaching plastics. However, beaching results are highly dependent on the beaching probability. On one hand, the beaching period varies from 3 years (with high probability) to 10 years (low probability). On the other hand, connectivity matrices show that beached particles mainly originate from the same country (for high probability) and from multiple different countries (for low probability). I find this analysis very interesting, especially to show our current limitations to properly simulate with numerical models the complex process of the beaching. I think that it is important to highlight the uncertainty in the beached patterns obtained and the relevance of improving the simulation of beaching in numerical models to obtain more robust results.

**SPECIFIC COMMENTS:**

\* In section 2.2.1 (Long-term simulations) the authors explain that they include the monthly variation of plastic waste input from rivers by releasing particles on the first day of every month. However, it is not clear the number of particles used in the simulation and the release location:

- Please specify if the points displayed in Figure 2a are the numerical release points. If that is the case, please include this information in the label of Figure 2.

- Please specify the number of particles release the first day of every month, the total number of particles and the initial spatial distribution of the particles. Is it the same for section 2.2.2?
\* Regarding the beaching probability (p), section 2.3 indicates that the beaching probability can vary between 0 and 1 and section 2.3.1 indicates that only 3 values of p are used. Please, clarify it.

\* L175 and Figure 3. Why the results shown in Figure 3 are without beaching?. I wonder if it would be more appropriate to show the results with beaching. Without beaching the transport between the different regions and the 'escape' mechanism may be overestimated. Please, provide more details about it.

\* As previously mentioned, I suggest to highlight (in the discussion and conclusions) the uncertainty in the beached patterns obtained related to the uncertainty in the beaching methodology.

---

## Author Comment (AC1) · 9 Sep 2020

Thank you for your kind comments. Thank you also for pointing out the typos, we have corrected these in the new version of the manuscript.

---

## Author Comment (AC2) · 9 Sep 2020

**Response to reviewer #2**

| | Reviewer comment | Author response |
|---|---|---|
| | The manuscript presents an interesting analysis of the transport and distribution of marine plastic debris from rivers into the northern Indian Ocean. The objective of the work is clear and the manuscript is well addressed and discussed. An interesting analysis is carried out to show how the results depend on the beaching methodology. The authors acknowledge that beaching of plastics is highly complex and that dynamics due to wind and waves are not considered in the simulations. These questions and their implications are identified and discussed in the manuscript. The manuscript represents a substantial contribution to scientific progress within the scope of Ocean Science and presents a high scientific and presentation quality.

However, I have some comments that I would like to be discussed in more detail: | Thank you. |
| 1 | 1. As the authors mention, they only consider the effect of surface currents on the transport of plastics in this study. This is correct, but this means that the results are representative of the marine plastic debris transported by surface currents. Buoyant items can be highly affected by wind, especially in coastal areas, where the wind can play an important role in the transport and beaching of marine debris. The authors would have to clarify the type of buoyant marine plastic debris under consideration and/or discuss in more detail how the results might change if the windage is included in the numerical simulations. | We explain why we have not considered the influence of wind and waves on the transport of beaching plastics in lines 301-311. In lines 314-318 we discuss the possible influence of including windage and/or Stokes drift on our results. We have clarified this in more detail by adding, lines 312-314: "Because we have not included wind and wave effects in our simulations, our results are likely applicable only to plastics that are neutrally or slightly positively buoyant and are transported in the upper 2 meters of the water column. Wind and waves can have a large influence on local beaching behaviour. However, on a large scale, …". |
| 2 | 2. One of the main objectives of the study is to determine which countries and islands are most heavily affected by beaching plastics. However, beaching results are highly dependent on the beaching probability. On one hand, the beaching period varies from 3 years (with high probability) to 10 years (low probability). On the other hand, connectivity matrices show that beached particles mainly originate from the same country (for high probability) and from multiple different countries (for low probability). I find this analysis very interesting, especially to show our current limitations to properly simulate with numerical models the complex process of the | This is partly already addressed in the Discussion in lines 284-290. We have emphasised this by adding to line 286: "it is therefore important to improve the simulation of beaching in numerical models and apply reliable beaching conditions".
We have also added an extra paragraph to the Discussion, lines 295-300: "In addition, we applied a single beaching probability throughout the Indian Ocean to our simulation results. Because beaching mechanisms depend on local coastal dynamics and morphology, beaching probabilities likely vary from location to location. A better understanding of the spatial |

| | | beaching. I think that it is important to highlight the uncertainty in the beached patterns obtained and the relevance of improving the simulation of beaching in numerical models to obtain more robust results. | variation of beaching probabilities depending on local conditions will likely improve the numerical simulation of beaching plastics. Finally, we did not take into account that beached plastics can also return to the ocean. Including these dynamics may also improve the simulation of beaching plastics. Recent works by Hinata et al. (2020b) and Hinata et al. (2020a) may contribute to this." |
|---|---|---|---|
| | SPECIFIC COMMENTS: | |
| **3** | 1. In section 2.2.1 (Long-term simulations) the authors explain that they include the monthly variation of plastic waste input from rivers by releasing particles on the first day of every month. However, it is not clear the number of particles used in the simulation and the release location:
• Please specify if the points displayed in Figure 2a are the numerical release points. If that is the case, please include this information in the label of Figure 2.
• Please specify the number of particles release the first day of every month, the total number of particles and the initial spatial distribution of the particles. Is it the same for section 2.2.2? | • The points displayed in Figure 2a show the release locations, except that some points may be shifted by one or two grid cells to prevent release of particles on land (these are minor shifts that are not visible on the scale shown in Figure 2a). This is described in section 2.2.1, lines 126-129: "… we release particles into the NIO from river plastic source locations (Figure 2a; Lebreton et al., 2017). Several of the source locations available from Lebreton et al. (2017) are located on land grid cells in HYCOM. We prevent releasing particles on or very close to land by increasing the HYCOM land mask with one grid cell and then moving any release locations on land to the nearest ocean grid cell (Figure A2)." We have clarified that the locations in Figure 2a are the particle release locations by adding in the caption of Figure 2a: "We release particles from these locations in our particle tracking simulations (section 2.2)."
• The number of particles that we release on the first day of every month is shown in Figure 2b (one particle represents 1 tonne of plastic waste). This is described in section 2.2.1, lines 129-131: "We include the monthly variation of plastic waste input from rivers (Figure 2b) in our simulation by releasing particles on the first day of every month. A single particle in our simulation represents 1 tonne of plastic waste." We have clarified this by adding in the caption of Figure 2b: "We release particles following this monthly variation in our particle tracking simulations, where 1 particle represents 1 tonne of plastic waste (section 2.2)." We have also added the total number of particles that are released in the simulation to line 131: "… we release a total |

| | | |
|---|---|---|
| | | of 267710 particles". The same release method is used in the monsoonal simulations described in section 2.2.2. We have clarified this by adding to line 143: "… using the same release method described in section …". |
| 4 | 2. Regarding the beaching probability (p), section 2.3 indicates that the beaching probability can vary between 0 and 1 and section 2.3.1 indicates that only 3 values of p are used. Please, clarify it. | We meant here that the beaching probability can be a value from 0 to 1; we were not referring to any specific values that we use in this study. We have clarified this by changing the sentence on lines 160-162: "The beaching probability can  assume values between a minimum value of 0 (no particles beach) and a maximum value of 1 (all particles within a distance $\Delta x$ of a coastline beach) per 5 days. |
| 5 | 3. L175 and Figure 3. Why the results shown in Figure 3 are without beaching? I wonder if it would be more appropriate to show the results with beaching. Without beaching the transport between the different regions and the 'escape' mechanism may be overestimated. Please, provide more details about it. | The purpose of this simulation was not to show the influence of beaching, but to illustrate how ocean surface currents influence the transport of particles in the NIO. These results are purely qualitative, meant to understand how the monsoon dynamics influence the transport of particles, and to understand how particles may cross from the NIO into the SIO. We do not quantify the transport between different regions in the NIO, or between the NIO and the SIO. Therefore, it is not an issue in this section that these transports may be overestimated because beaching is not included. We have clarified this by adding on lines 177-178: "We do not include any beaching effects in these simulation results, because our purpose with this simulation is to qualitatively illustrate the transport of particles by ocean surface currents." |
| 6 | 4. As previously mentioned, I suggest to highlight (in the discussion and conclusions) the uncertainty in the beached patterns obtained related to the uncertainty in the beaching methodology. | See our response to comment #2 above. |